# Boosting Text-to-Video Generative Model with MLLMs Feedback

**Xun Wu**[1], **Shaohan Huang**[1✉], **Guolong Wang**[2], **Jing Xiong**[3], **Furu Wei**[1]

[1] Microsoft Research Asia, [2] University of International Business and Economics
[3] The University of Hong Kong

xunwu@microsoft.com, shaohanh@microsoft.com, fuwei@microsoft.com

## Abstract

Recent advancements in text-to-video generative models, such as Sora [3], have showcased impressive capabilities. These models have attracted significant interest for their potential applications. However, they often rely on extensive datasets of variable quality, which can result in generated videos that lack aesthetic appeal and do not accurately reflect the input text prompts. A promising approach to mitigate these issues is to leverage Reinforcement Learning from Human Feedback (RLHF), which aims to align the outputs of text-to-video models with human preferences. However, the considerable costs associated with manual annotation have led to a scarcity of comprehensive preference datasets. In response to this challenge, our study begins by investigating the efficacy of Multimodal Large Language Models (MLLMs) generated annotations in capturing video preferences, discovering a high degree of concordance with human judgments. Building upon this finding, we utilize MLLMs to perform fine-grained video preference annotations across two dimensions, resulting in the creation of VIDEOPREFER, which includes 135,000 preference annotations. Utilizing this dataset, we introduce VIDEORM, the first general-purpose reward model tailored for video preference in the text-to-video domain. Our comprehensive experiments confirm the effectiveness of both VIDEO-PREFER and VIDEORM, representing a significant step forward in the field.

## 1  Introduction

Diffusion models have significantly enhanced the quality of generation across various media formats, including images [30, 31, 55] and videos [13, 22, 42]. In the context of text-to-video generation, recent advancements in text-to-video diffusion models, exemplified by Sora [4], have achieved remarkable success in generating high-quality videos from textual prompts. However, despite recent progress, the visual fidelity of generated videos still presents opportunities for refinement [38]. Video generation poses greater challenges than image generation, as it entails modeling a higher-dimensional spatio-temporal output space while remaining conditioned solely on a textual prompt. Consequently, existing text-to-video generative models often produce results that are visually unappealing and inadequately aligned with the provided textual prompts.

To address these limitations, a pivotal solution is Reinforcement Learning from Human Feedback (RLHF), which has proven its efficacy in text-to-image diffusion models [51, 2, 26]. These methods aim to fine-tune a diffusion model to maximize a reward function corresponding to specific aspects of image quality or alignment with text prompts. *Despite this, when a Reward Model trained in the image domain is transferred to the video domain, it can exhibit significant disparities with the objectives of video optimization (e.g., InstructVideo [53]).* Training a reward model directly in the video domain is becoming increasingly urgent, but *the scale of preference datasets in the video field is small* (see in Table 1).

38th Conference on Neural Information Processing Systems (NeurIPS 2024).

Table 1: Statistics of existing preference datasets for text-to-video generative models. * denotes annotated by human while † denotes annotated by GPT-4 V.

| Dataset | Prompts | Videos | Preference Choices |
|---|---|---|---|
| VBench [15]* | 1K | 4K | 44K |
| TVGE [56]* | 0.5K | 2.5K | 2.5K |
| T2VQA-DB [18]* | 1K | 45K | 45K |
| VIDEOPREFER† | **14K** | **54K** | **135K** |

Table 2: Correlations between MLLMs and human judgment on text-video alignment on TVGE datasets [56]. * denotes image-based MLLMs while † denotes Video-based MLLMs.

| Model | Spearman $\rho$ | Kendall $\tau$ | Acc (%) |
|---|---|---|---|
| Video-LLaMA [54]† | 0.288 | 0.206 | – |
| mPLUG-OWL2-V [52]† | 0.394 | 0.285 | 61.87 |
| InstructBLIP [10]* | 0.342 | 0.246 | 54.33 |
| mPLUG-OWL2-I [52]* | 0.358 | 0.257 | 53.36 |
| Gemini pro Vision* | 0.3921 | 0.2993 | 64.71 |
| LLaVA 1.6-34B* | 0.3139 | 0.2278 | 53.20 |
| GPT-4 V* | **0.486** | **0.360** | **69.65** |

Recognizing the importance of addressing these challenges in text-to-video generative models, we first construct a large-scale fine-grained preference benchmark by utilizing MLLMs as annotators, namely VIDEOPREFER. VIDEOPREFER contains following strength: (1) VIDEOPREFER is the largest open-source video preference dataset, containing 135,000 preference choices (see in Table 1). (2) Different from existing video preference datasets, VIDEOPREFER contains true video captured by human, making VIDEOPREFER more generalizable. (3) VIDEOPREFER is annotated using Multimodal Large Language Models, which is cost-effective and easily scalable.

Based on VIDEOPREFER, we release the first general-purpose text-to-video preference reward model, VIDEORM. Unlike the preference rewards in the image domain used by previous methods for video generation alignment [53], our VIDEORM automatically captures the temporal information in videos, enhancing the modeling of quality assessment and alignment. Furthermore, we investigate the alignment of text-to-video generative models using VIDEORM and conduct extensive experiments that demonstrate its efficacy as a reward model and metric for video alignment. This significantly improves the generation quality of video generation models across multiple aspects. Our main contributions are:

- We systematically identify the challenges for text-to-video human preference annotation. Consequently, we employ MLLMs for preference annotation and construct the largest and most comprehensive video preference dataset, **VIDEOPREFER**.
- We systematically identify the lack of general-purpose text-to-video preference reward model and propose **VIDEORM**, outperforming existing reward models for text-to-video alignment.
- Extensive experimental results validate the effectiveness of both the VIDEOPREFER and VIDEORM. Furthermore, through detailed discussions, we demonstrate that MLLMs are effective and cost-effective as annotators for video preferences, revealing the promising future prospects of RLAIF in the domain of video preference alignment.

## 2 VIDEOPREFER

We introduce VIDEOPREFER, a large-scale, fine-grained video preference dataset constructed by collecting feedback from multimodal large language model annotators. In total, VIDEOPREFER contains 135K pairs of binary preference choices for 54K videos. In § 2.1, we provide evidence demonstrating that GPT-4 V is a human-aligned preference annotator in the video domain. The construction pipeline of VIDEOPREFER is introduced in § 2.2. Detailed analysis of the statistics of VIDEOPREFER can be found in Appendix § B.1.

### 2.1 Why GPT-4 V(ision) can act as a human-aligned preference annotator?

GPT-4 V has already demonstrated annotation performance aligned with human consistency in text-to-image generation [7, 46, 44]. To further validate GPT-4 V's ability to provide reliable preference annotations in the video domain, we used different MLLMs to annotate a subset of the TVGE dataset and calculated their correlation with the ground truth annotations, e.g., accuracy and kendall. The results are shown in Table 2. We find that (1) GPT-4 V's annotation accuracy and correlation are the best among all MLLMs. (2) The accuracy of GPT-4 V is 69.65%, which is very close to the previously reported agreement rate between qualified human annotators (approximately 70% [9, 46, 25, 11]). This demonstrates that GPT-4 V is a reliable annotator.

## 2.2 Construction Pipeline

The construction pipeline of VIDEOPREFER can be split into following three steps:

**Step-1: Prompts Collection.** We collect prompts from VidProM [40] which contains 1.67 million unique text-to-video prompts, as well as from two video-captioning benchmarks: ActivityNet[5] and MSR-VTT[49]. For VidProM, we just randomly sample 12.9k prompts from its corpus. For video-captioning benchmarks, we directly utilize the provided video segment captions from the original dataset as prompts. For instance, in the ActivityNet [5], we utilize the text provided in caption corpus, comprising 0.9K thousand text instances corresponding to distinct video segments. Consequently, we obtain a total of 14K prompts, constituting the prompt set for our VIDEOPREFER.

**Step-2: Video Collection & Generation.** We generate videos with text-to-video models and collect real-world videos based on prompts collected in Step-1 for preference annotation. Through these two ways, we obtained 54K video candidates:

· *Model-generated videos.* We selected the top-ranked text-to-video generation models on Hugging-Face[1] (e.g., ModelScopeT2V [38]), as well as the open-source state-of-the-art text-to-video models (e.g., Open-Sora [57]), to constitute our video generation model pool. Details regarding these models, as well as the proportions and resolutions of the generated videos, can be found in Table 5. For each prompt, we randomly sample models from the model pools, along with class-free guidance scales, to generate four different corresponding videos. This approach results in a high degree of diversity, facilitating the training of a more generalizable and comprehensive preference reward model.

· *Real-world videos.* Given the substantial quality disparity between videos generated by existing text-to-video generation algorithms and real videos, we also incorporate real-world videos to enhance the generalizability and diversity of our VIDEOPREFER. Specifically, we incorporate video segments from the two datasets introduced in Step-1 (ActivityNet [5] and MSR-VTT [49]), as one of the four candidate videos, with the remaining three being generated as above.

**Step-3: Preference Generation.** Due to the prohibitively high cost of manual annotation, we utilize the state-of-the-art multimodal large language models (MLLMs), i.e., GPT-4 V to annotate video preferences, enabling extensive and fine-grained annotation at scale. The reliability of GPT-4 V as annotators is thoroughly discussed in § 2.1. Specifically, for each prompt and its corresponding four video candidates, we employ GPT-4 V to assign preference scores on a 1-to-5 Likert scale to each video candidate for two aspects: *Prompt-Following* and *Video Quality*. Lower scores indicate lower preference and vice versa. Finally, we obtain 135K preference choices. Detailed input instructions used for GPT-4 V's annotation can be found in the Appendix E.

By performing the above three steps, we finally have 14K data items in VIDEOPREFER, while each data item contains a textual prompt and four corresponding generated videos. For each video, there are preference annotations for two aspects: Prompt-Following and Video-Quality provided. More details of VIDEOPREFER, e.g., visualization of example data item (see in Figure 10), can be found in Appendix B.

## 3 VIDEORM: A General-Purpose Video Preference Reward Model

Based on VIDEOPREFER, we implement the reward model training and derive the first general-purpose video preference reward model, VIDEORM. We detail the architecture of VIDEORM in § 3.1, its optimization objectives in § 3.2, and the methodology for fine-tuning video generative models with VIDEORM in § 3.3.

### 3.1 Architecture

Existing text-to-video alignment work [53] adopts HPS v2 [45] as the reward model, which is fine-tuned from CLIP [27] and optimized upon a large image preference benchmark and demonstrates state-of-the-art preference evaluation capabilities for text-to-image generation.

However, we contend that the direct application of HPS v2 may lead to a one-sided assessment of video preferences, as it lacks the capability to evaluate the overall attributes of a video, such

---

[1]https://huggingface.co/models?pipeline_tag=text-to-video

as temporal coherence and dynamics. Nevertheless, given the strong correlation between video preferences and the preferences of individual frames, the proficiency of HPS v2 [45] in evaluating single-frame image preferences can still aid in enhancing the assessment of video preferences. Thus, to achieve a better evaluation for video preferences, we modify the structure of HPS v2 by integrating several temporal modeling modules, to develop a specialized reward model VIDEORM for the video domain.

The full architecture of VIDEORM is shown in Figure 1. Specifically, inspired by recent advancements [39, 21] in enhancing the temporal modeling capabilities of the CLIP [27], we add two kinds of temporal modeling modules into HPS v2 to modeling videos:

**Temporal Shift** [20], a parameter-free module which shifts part of the feature channels along the temporal dimension and facilitates information exchange between neighboring input frames. Following [39], we insert the module between every two ViT Layers of the image encoder of HPS v2.

**Temporal Transformer**. The sequence of frame-wise features extracted by the image encoder are then fed into a temporal transformer to modeling the temporal features of the video.

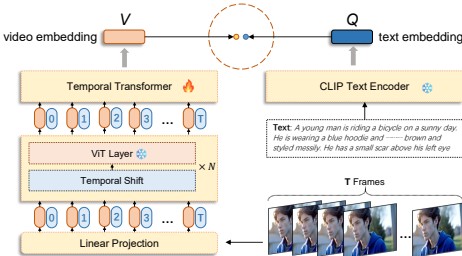

Figure 1: **Overview of VIDEORM**. By incorporating temporal modeling modules, VIDEORM is capable of not only capturing the preference scores of individual frames but also modeling the temporal features of the video, thereby better evaluating the overall preference score of the video.

## 3.2 Optimization

During the optimization process of VIDEORM, we freeze the text encoder and ViT layers contained in the image encoder to retain the single-frame preference modeling capability of HPS v2, while only optimizing the temporal transformer module. Thus, VIDEORM achieves the capability to model video preferences from both the perspectives of individual frames and temporal dynamics, allowing for a more comprehensive understanding of video content.

Similar to previous works [36, 25], to train VIDEORM on VIDEOPREFER, we first average the scores of all 16 aspects for each video candidate, obtaining the final preference score for that video candidate. Then we formulate the preference score for each pair-wise video candidates as rankings. Thus, for each data item (consisting of one prompt $\mathbf{T}$ and its corresponding four video candidates $\mathbf{v}_1, \mathbf{v}_2, \mathbf{v}_3$ and $\mathbf{v}_4$), we get at most $C_4^2$ comparison pairs if there are no ties between two video candidates. For each comparison, if $\mathbf{v}^+$ is better and $\mathbf{v}^-$ is worse ($\mathbf{v}^+ \succ \mathbf{v}^-$), the loss function can be formulated as:

$$\text{loss}(\theta) = -\mathbb{E}_{(\mathbf{T}, \mathbf{v}^+, \mathbf{v}^-) \sim \mathcal{D}} \left[ \log \left( \sigma \left( R_\theta \left( \mathbf{T}, \mathbf{v}^+ \right) - R_\theta \left( \mathbf{T}, \mathbf{v}^- \right) \right) \right) \right] \tag{1}$$

where $R_\theta(\mathbf{T}, \mathbf{v})$ is a scalar value of preference model for prompt $\mathbf{T}$ and generated video $\mathbf{v}$.

## 3.3 Fine-tuning Text-to-Video Models with VIDEORM

Current exploration of reward reinforcement learning algorithms for fine-tuning text-to-video generative models is quite limited. The only related work is InstructVideo [53], which utilizes an image-domain reward model (HPS v2 [45]) to fine-tune text-to-video models.

Consequently, InstructVideo may have the following shortcomings: (1) Due to the inherent gap between images and videos, directly using image-domain reward models cannot accurately calculate the reward for generated videos, leading to visual artifacts such as structural twitching and color jittering [53]. (2) Additionally, InstructVideo [53] calculates the reward value for all frames selected from generated videos throughout the full DDIM [34] sampling procedure, making the fine-tuning process highly sample inefficient.

VIDEORM is specifically designed for the video domain. It is initialized with the weights of the best image-domain reward model and trained on video preference datasets, utilizing temporal modeling mechanisms (e.g., temporal transformers) to capture temporal features of videos. This enables it to evaluate the reward score of generated videos more effectively from both individual frame and

temporal perspectives. Additionally, unlike the image-domain reward model used in InstructVideo, which takes single video frames as input, VIDEORM processes the entire video as input. Therefore, our approach is more computational efficient and can directly leverage various effective image-domain reward reinforcement learning algorithms (such as DRaFT [8]) without the need to balance rewards between frames generated at each step, as required by InstructVideo.

Based on the above, we integrated VIDEORM into the image-domain DRaFT [8] algorithm and design a reward reinforcement learning algorithm suitable for text-to-video generative models, named DRaFT-V. The specific algorithm details are shown in Algorithm 1. DRaFT-V truncates the backward pass, differentiating and computing reward score from VIDEORM through only the last $K$ steps, making the full fine-tuning process more efficient.

In implementation, rather than fine-tuning the full set of model parameters, we follow [53] to adopt LoRA [14] to further accelerate fine-tuning and circumvent the issue of computational intensity as well as the risk of catastrophic forgetting associated with the reward loss in diffusion models.

---

**Algorithm 1** DRaFT-V: Reward Reinforcement Learning for Fine-tuning Text-to-Video Models with VIDEORM

1: **Dataset:** Prompt set $\mathcal{Y} = \{y_1, y_2, ..., y_n\}$
2: **Pre-training Dataset:** Text-Video pairs dataset $\mathcal{D} = \{(t_1, v_1), ...(t_n, v_n)\}$
3: **Input:** Text-to-video models $\Upsilon$ with pre-trained parameters $\theta_0$, VIDEORM $R$, reward-to-loss map function $\theta$, Text-to-video models pre-training loss function $\psi$, reward re-weight scale $\lambda$
4: **Initialization:** The number of noise scheduler time steps $T$, and the truncating step $K$
5: **for** $y_i \in \mathcal{Y}$ and $(t_i, v_i) \in \mathcal{D}$ **do**
6:    $\mathcal{L}_{pre} \leftarrow \psi_{\theta_i}(t_i, v_i)$
7:    $w_i \leftarrow w_i$ // Update $\Upsilon_{\theta_i}$ using $\mathcal{L}_{pre}$
8:    $x_T \sim \mathcal{N}(0, I)$ // Sample noise as latent
9:    **for** $j = T, ..., 1$ **do**
10:      **if** $j > K$ **then**
11:        **no grad:** $x_{j-1} \leftarrow \Upsilon_{\theta_i}\{x_j\}$
12:      **else**
13:        **with grad:** $x_{j-1} \leftarrow \Upsilon_{\theta_i}\{x_j\}$
14:        $x_0 \leftarrow x_j$ // Predict the original latent by noise scheduler
15:        $z_i^j \leftarrow x_0$ // From latent to image
16:        $\mathcal{L}_{reward} \leftarrow \lambda\phi(r(y_i, z_i^j))$ // Reward loss
17:      **end if**
18:    **end for**
19:    $\theta_{i+1} \leftarrow \theta_i$ // Update $\Upsilon_{\theta_i}$ using $\mathcal{L}_{reward}$
20: **end for**

---

# 4 Experiments

We conduct extensive experiments to validate the effectiveness of VIDEOPREFER and VIDEORM. We first train VIDEORM and evaluate it on existing human-preference benchmark (§ 4.1). Next, we fine-tune existing text-to-video diffusion models for aligning human preference by utilizing DRaFT-V with VIDEORM (§ 4.2). Finally, we present ablation studies (§ 4.3).

## 4.1 Task 1: Reward Modeling

**Setup**. Based on VIDEOPREFER, we develop VIDEORM, an advanced open-source general-purpose reward model that provides preferences for generated videos. Specifically, we train three versions of VIDEORM. (1) Firstly, to validate the effectiveness of VIDEOPREFER, we average the preference scores in each aspect of VIDEOPREFER to get a final preference score, and train **VIDEORM-V** with the merging version of VIDEOPREFER. (2) Then, we train **VIDEORM-H** on several open-source human-crafted preference datasets listed in Table 1. (3) Finally, to build a stronger RM for text-to-video generation, we mix these open-source human-crafted preference datasets with VIDEOPREFER to train **VIDEORM**. The details for dataset processing can be found in Appendix A. All VIDEORM series models are trained in half-precision on $8 \times 32$GB NVIDIA V100 GPUs, with a learning rate of 1e-5 and batch size of 64 in total. We set the input frames $N = 8$.

**Compared Baselines**. Due to the lack of reward models in the video domain, we compare VIDEORM with state-of-the-art reward models from the image domain, i.e., CLIP ViT-H/14 [27], ImageReward [48], PickScore [17] and HPS v2 [45]. For all these reward models from the image domain, we calculate the scores for all video frames and then take the average as the final reward score for the video.

**Preference Accuracy**. The preference prediction accuracy across test benchmarks from three human-crafted preference datasets are reported in Table 3. As we can see, the VIDEORM series outperform baseline reward models by a large margin, indicating that VIDEORM series are the best open-source reward models for text-to-video domain. We also find that VIDEORM-V which does not train on any open-source video preference datasets also surpasses all other baselines. This result validates the high quality of VIDEOPREFER enables strong out-of-distribution generalization and validate

Table 3: Pair-wise preference prediction accuracy across human-crafted preference datasets. The Aesthetic Classifier (simplified as Aesthetic) makes prediction without seeing the text prompt. The best results are in blod and the second are underlined.

| Model | TVGE [56] | VBench [15] | T2QA-DB [18] | Avg |
|---|---|---|---|---|
| CLIP ViT-H/14 [27] | 57.3 | 52.7 | 52.1 | 54.0 |
| Aesthetic [33] | 56.1 | 51.0 | 52.7 | 53.3 |
| ImageReward [48] | 66.8 | 54.3 | 53.9 | 58.3 |
| PickScore [17] | 64.7 | 53.9 | 61.2 | 59.9 |
| HPS v2 [45] | 69.5 | 55.7 | 52.8 | 59.3 |
| VIDEORM-H | 72.8 | 60.2 | 64.3 | 65.8 |
| VIDEORM-V | 73.0 | 61.1 | 65.1 | 66.4 |
| VIDEORM | **73.7** | **63.5** | **65.4** | **67.5** |

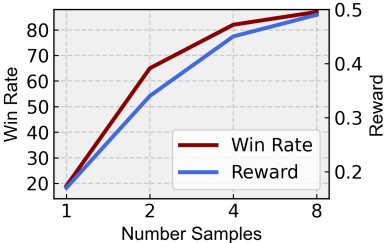

Figure 2: Best-of-$n$ experiments on the T2VQA-DB [18] test benchmark. We sample $n$ generated videos and choose the one with the highest reward.

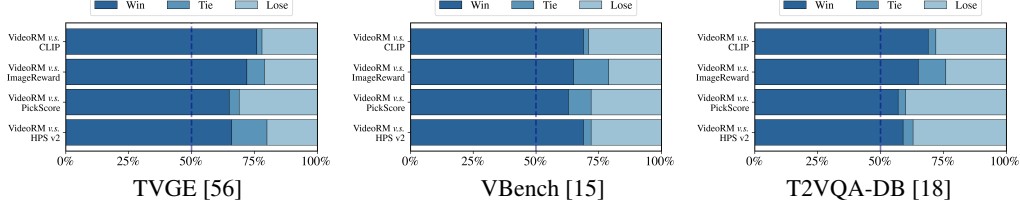

TVGE [56]          VBench [15]          T2VQA-DB [18]

Figure 3: Win rates of VIDEORM compared to other reward models across three test benchmarks. On average, 72% to CLIP, 67% to ImageReward, 62% to PickScore and 65% to HPS v2.

the effectiveness of treating state-of-the-art MLLMs, i.e., GPT-4 V, as preference annotator for text-to-video generation domain.

**Best-of-$n$ Experiments**. To verify that our VIDEORM could serve as a good indicator of video generation quality, we conduct best-of-$n$ experiments on the T2VQA-DB [18] test benchmark. For each data item, which includes a prompt and ten corresponding videos, we calculate the reward score for each of the ten videos with VIDEORM. Thereafter, we select the best-of-$\{1, 2, 4, 8\}$ responses and calculate their scores. The final results are presented at Figure 2, we find the win rate at the test benchmark increases proportionally with rewards, which validates that our VIDEORM gives rigorous rewards that reflect the overall preference score.

**Human Evaluation**. To evaluate the ability of VIDEORM to select the more preferred videos among large amounts of generated videos, we produce human study. We randomly select 200 textual prompts from these three test benchmarks and generate 32 different videos for each prompt by utilizing ModelScopeT2V [38]. Then, we perform different reward to select from those videos to get top3 results. After that, five annotators are asked to identify which video was superior or if both were of equal quality (denoted as 'Tie') and we show the corresponding win rates against other reward models [45] at Figure 3. Qualitative results can be seen at Figure 11 in Appendix. All of these results show that VIDEORM can select videos that are more aligned to text and with higher fidelity and avoid toxic contents.

### 4.2   Task 2: Fine-tuning Text-to-Video Generative Models

**Setup.** Based on VIDEORM, we validate the effectiveness of DRaFT-V. We adopt the publicly available text-to-video diffusion model ModelScopeT2V [38] as base model, which is trained on WebVid10M with T = 1000 and is able to generate videos of $16 \times 256 \times 256$ resolution. We random sample 2K prompt-video pair from the mixture of existing video preference datasets (TVGE [56], VBench [15] and T2VQA-DB [18]) and VIDEOPREFER as the training data. Each group of experiment is trained in half-precision on $4 \times 32$GB NVIDIA V100 GPUs, with a learning rate of 1e-5, batch size of 8, fine-tuning step of 400 and $K$ adopted in DRaFT-V as 10 in total.

**Compared Baselines.** Since the only existing reward-based fine-tuning algorithm for text-to-video models is InstructVideo [53]. We compare our DRaFT-V with the the baseline text-to-video model (i.e., ModelScopeT2V [38]) and InstructVideo implemented by ourselves. Note that in the aforementioned algorithms, InstructVideo uses HPS v2 [45] as the reward model. Additionally, to verify the effectiveness of using VIDEORM as a reward model for fine-tuning video generation models, we



TVGE [56]    VBench [15]    T2VQA-DB [18]

Figure 5: Win rates of text-to-video models fine-tuned with DRaFT-V compared to other baselines. Here baseline denotes the base text-to-video model without any fine-tuning.

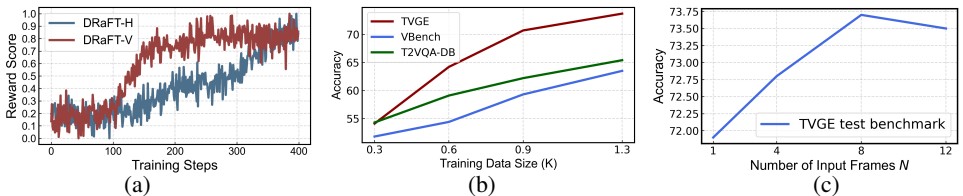

(a)    (b)    (c)

Figure 6: (a) Visualization of reward model values change with the training steps for both DRaFT-V and DRaFT-H. (b) Evaluation result across three test benchmarks for the size of training data used in optimizing VIDEORM. (c) Ablation study for the number of input frames $N$ in VIDEORM.

introduced two comparison groups: (1) DRaFT-H, which uses the same algorithm as our DRaFT-V (i.e., DRaFT [8]), but with HPS v2 as the reward model. (2) InstructVideo-V, which uses the same algorithm as InstructVideo [53], but with VIDEORM as the reward model.

**Generation Quality Evaluated by Multiple Reward Models.** At first, we visualize the evolution of model's generation quality across multiple reward models as the model training steps increase when using our DRaFT-V at Figure 4. We observe that with the progress of training, all reward models show an increasing trend, indicating that VIDEORM can serve as a reliable reward model to enable the generated videos to align more closely with human preferences.

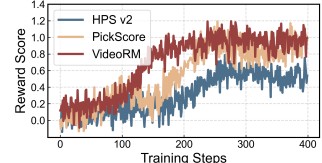

Figure 4: Evaluation results of the text-to-video model's generation quality across multiple reward models when maximizing scores from VIDEORM during the DRaFT-V fine-tuning process.

**Human Evaluation.** To evaluate the generative ability of text-to-video models fine-tuned with DRaFT-V and other compared baselines, we produce human study. We randomly select 100 prompts from these three test benchmarks (TVGE [56], VBench [15] and T2VQA-DB [18]), and for each prompt, we generate videos using different fine-tuned models. Five annotators are asked to identify which video was superior or if both were of equal quality (denoted as 'Tie') and we show the corresponding win rates at Figure 5. We find that: (1) All of these results show that DRaFT-V can generate videos that are more prefered by human. (2) By comparing DRaFT-V with DRaFT-H, as well as InstructVideo-V with InstructVideo, we demonstrate that VIDEORM is a more effective reward model for fine-tuning text-to-video models.

Besides, qualitative results are presented at Figure 12 in Appendix D. We find that: (1) The base generation model often fails to align videos with prompt descriptions, but fine-tuning with a reward model (image or video domain) improves quality and prompt consistency, demonstrating the effectiveness of reward-based fine-tuning. (2) Fine-tuning with VIDEORM (DRaFT-V and InstructVideo-V) significantly outperforms fine-tuning with an image domain reward model HPS v2, indicating VIDEORM's superiority for text-to-video model fine-tuning.

**Efficiency Evaluation.** We compare the efficiency of our algorithm on two aspects: (1) Convergence speed: we visualize the changes in reward values with training steps for DRaFT-H and DRaFT-V in Figure 6 (a). We find that under the same reward-based fine-tuning algorithm, using VIDEORM leads to earlier convergence and better performance (as demonstrated by the results above) compared to HPS v2, further validating the effectiveness of VIDEORM in fine-tuning text-to-video models. (2) Inference speed. For a single video with 8 frames, HPS v2 requires approximately an average of 5 seconds for inference, while VIDEORM only needs an average of 1.3 seconds. Besides, InstructVideo requires 20 DDIM steps for a single fine-tuning step, taking approximately an average of 8.8 seconds, while DRaFT-V only needs an average of 2.3 seconds.

### 4.3 Ablation Study

**Scalability.** To investigate the effect of training dataset sizes on the performance of VIDEORM, and to verify the scalability of VIDEOPREFER, comparative experiments are conducted. Note that VIDEORM in this experiment are only trained on VIDEOPREFER. Figure 6 (b) shows that adding up the scale of the dataset significantly improves the preference accuracy of VIDEORM. It's promising that if we employ GPT-4 V to collect more annotation data in the future, VIDEORM will get better performance.

**Number of Video Frames $N$ Input to VIDEORM.** We investigate the impact of different numbers of input video frames on the performance of the RM model. Specifically, we set the number of input frames to $N$ = 1, 4, 8, and 12 and test corresponding model at TVGE [56] test benchmark. The corresponding results are presented at Figure 6 (c), we find that the VIDEORM's performance significantly decreases when $N$ is small. This is promising because too few input frames prevent the model from accurately capturing all the information and temporal features of the video, e.g., the degree of motion and temporal consistency.

**Temporal Feature Modeling method in VIDEORM.** We conduct experiments to explore the impact of different video frame temporal feature modeling methods on the performance of the reward model. In VIDEORM, we use a temporal transformer trained from scratch to model the temporal features of video frames. Additionally, we conducted ablation experiments by replacing this transformer with 1D Convolutional Layer (denoted as VIDEORM$^\dagger$) and LSTM (denoted as VIDEORM$^\ddagger$). Note that we adjusted the number of layers in the temporal feature modeling module to ensure that the number of trainable parameters remains equal. Besides, Temporal Shift are

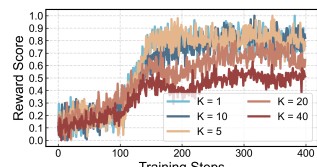

Figure 7: Ablation over $K$ adopted in DRaFT-V during fine-tuning text-to-video model.

employed for all compared methods. The results are presented at Table 4. We find that: (1) All models outperform the image-domain reward model. (2) Replacing the temporal transformer with LSTM achieved comparable performance, indicating that VIDEORM is robust to the choice of model architecture for temporal feature modeling. (3) Replacing the temporal transformer with Conv1D resulted in a significant performance drop. We hypothesize that this is because Conv1D cannot effectively model temporal features, leading to VIDEORM$^\dagger$'s inability to accurately evaluate video generation quality from a temporal perspective. This highlights the effectiveness of using a temporal transformer for modeling temporal features in VIDEORM.

**Backbone of VIDEORM.** We explore the impact of different backbone on the final performance of VIDEORM. In our design, VIDEORM is initialized with the weights of HPS v2 [45], which is demonstrated as the best reward model for image domain. Here we replace HPS v2 with other reward models in image domain, e.g., PickScore [17] (denoted as VIDEORM$^\sigma$) and ImageReward [48] (denoted as VIDEORM$^\beta$) while keep the count of trainable parameter equal. The corresponding results are presented at Table 4. We find that the final performance of the corresponding VIDEORM is proportional to the performance of the initialized backbone weights. For example, ImageReward, which has the lowest performance among the three (see Table 3), corresponds to the worst VIDEORM$^\beta$, while PickScore, which has the best performance among the three (see Table 3), corresponds to the best VIDEORM$^\sigma$.

**Step $K$ adopted in DRaFT-V.** We investigate the impact of different $K$ adopted in DRaFT-V on the final generative performance. The results are presented at Figure 7, we find that: (1) Performance degrades as K increases for $K$ > 10. (2) Even with smaller values of $K$, e.g., $K$ = 1, DRaFT-V can achieve good results. Both findings are aligned with the results in DRaFT [8].

## 5 Analysis

In this section, we analyze the impact of different parameter selections—specifically, the number of frames ($N$) sampled from a single video and the temperature setting $\tau$ of the GPT-4 V, on the accuracy of GPT-4 V-generated annotations. Due to GPT-4 V's limitation of processing a maximum of 10 frames, we analyze annotation accuracy for $N$ = 4, 6, 8, and 10 frames. The temperature $\tau$ controls the diversity and randomness of GPT-4 V's outputs. We analyze annotation accuracy for $\tau$ = 0.3, 0.5, 0.7, and 0.9.

Table 4: Ablation study for VIDEORM. The Aesthetic Classifier (simplified as Aesthetic) makes prediction without seeing the text prompt.

| Model | TVGE [56] | VBench [15] | T2VQA-DB [18] | Avg |
|---|---|---|---|---|
| VIDEORM$^\dagger$ | 70.1 | 58.4 | 61.5 | 63.3 |
| VIDEORM$^\ddagger$ | **73.9** | 62.7 | 65.2 | 67.3 |
| VIDEORM$^\sigma$ | 72.2 | **64.2** | **66.9** | 67.7 |
| VIDEORM$^\beta$ | 72.4 | 63.6 | 64.2 | 66.7 |
| VIDEORM | 73.7 | 63.5 | 65.4 | **67.5** |

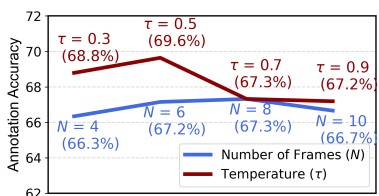

Figure 8: The impact of different hyperparameter choices on the annotation accuracy of GPT-4 V.

The summarized results are shown in Figure 8. For $N$, accuracy initially increases with more frames but decreases beyond $N = 8$, achieving the highest accuracy at $N = 8$. This is because more frames allow for a comprehensive evaluation, but too many frames can lead to overly long contexts, reducing accuracy due to GPT-4 V's limited capacity. For $\tau$, lower values yield higher accuracy, likely because reduced randomness leads to more comprehensive predictions, while higher randomness might cause GPT-4 V to overly focus on specific frames or aspects, resulting in less comprehensive outcomes.

## 6 Related Work

Generative models, particularly those based on diffusion techniques, have demonstrated high-quality generation capabilities [32, 28, 31, 30, 35] by training on extensive internet-scale datasets, but the mixed quality of these datasets often leads to visually unappealing and misaligned outputs.

**Aligning text-to-image generative models.** Aligning text-to-image generative models [37, 17, 8, 26, 37] has garnered increasing attention in recent years and has shown promising results in producing outputs that are more aligned with human preference. Imagereward [48], HPS v2 [47, 45], and PickScore [17] are the three most commonly used reward models for align text-to-image models. They are respectively trained on three major image domain preference datasets, namely Imagereward, ImageRewardDB [48], HPD [47, 45] and Pick-a-Pic [17]. The effectiveness of these reward models has been validated across alignment algorithmss like AlignProp [26], DRAFT [8] and ReFL [48].

**Aligning text-to-video generative models.** Compared to text-to-image, exploration related to aligning text-to-video generative models is relatively sparse. InstructVideo [53] instruct text-to-video diffusion models with by fine-tuning with existing image-wise preference reward model HPS v2. We hypothesize that this may restrict effective preference evaluations for generated videos, as image-based reward models cannot adequately capture temporal features, impairing assessments of video coherence and dynamics. Additionally, preference datasets for text-to-video generation are also scarce. The lack of large-scale and effective open-source preference datasets severely restricts the development of research related to align text-to-video generative models.

**Reinforcement Learning from AI Feedback.** LLMs have been extensively used for data generation [43, 24], augmentation [12] and in self-training setups [41, 23]. Some works [1] introduced the idea of reinforcement learning from AI feedback (RLAIF), which used LLMs labeled preferences in conjunction with human labeled preferences to jointly optimize for the two objectives of helpfulness and harmlessness. Recent works have also explored related techniques for generating rewards from LLMs [29, 19, 50]. These works demonstrate that LLMs can generate useful signals for reinforcement learning fine-tuning. However, leveraging the feedback from MLLMs for aligning text-to-image generative model or for text-to-video generative model is less explored.

## 7 Conclusion

In this paper, we identify the current obstacles faced in aligning text-to-video generative models, i.e., lacking of large-scale preference datasets and reward models specifically tailored for videos. Thus, we introduce VIDEOPREFER and VIDEORM to address the aforementioned issues. Experimental validation confirms that GPT-4 V can act as a human-aligned preference annotator. We utilized it to label 135K video preference annotation, forming VIDEOPREFER. Based on this, we trained a video-specific reward model, VIDEORM. Extensive analytical usage has demonstrated the effectiveness of both VIDEOPREFER and VIDEORM.

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

# A Datasets Processing

**T2VQA-DB [18]**. The T2VQA-DB dataset only comprises a single overall preference evaluation score for each data item. We randomly selected 2.3K samples to serve as the final test benchmark.

**TVGE [56]**. The TVGE dataset comprises preference evaluations from 2 aspects, namely: *text alignment* and *video quality*. We just simply average the scores across these two aspects to obtain a final overall evaluation score as with the aforementioned datasets. We randomly selected 15K samples to serve as the final test benchmark.

**VBench [15]**. The VBench dataset comprises preference evaluations from 16 aspects, namely: *subject consistency*, *background consistency*, *temporal flickering*, *motion smoothness*, *dynamic degree*, *aesthetic quality*, *imaging quality*, *object class*, *multiple objects*, *human action*, *color*, *spatial relationship*, *scene*, *temporal style*, *appearance style*, *overall consistency*.

Due to the fact that the videos and prompts evaluated for preference in each aspect are largely non-overlapping, we cannot simply average the scores across all aspects to obtain a final overall evaluation score as with the aforementioned datasets. Instead, we mixed all samples from the 16 aspects and randomly selected 10K samples to serve as the final test benchmark.

# B VIDEOPREFER

## B.1 Statistic of VIDEOPREFER

In Table 5, we present the sources and distribution of the videos in our VIDEOPREFER. We find that VIDEOPREFER comprises a diverse range of video sources, including videos generated by state-of-the-art text-to-video models as well as real videos. This extensive variety of video sources enhances the robustness and generalization capabilities of VIDEOPREFER.

We visualize the distribution of preference annotations across two evaluation aspects (prompt-following an video-quality) in VIDEOPREFER in Figure 9. We find that the overall score distribution of the dataset is close to the normal distribution for the Prompt-Following evaluation perspective. From the perspective of Video-quality evaluation, the overall distribution of data sets tends to be low, which indicates that the generation effect of the existing text-to-video model is not satisfactory.

Besides, we also visualize some data examples in VIDEOPREFER at Figure 10.

Table 5: Video sources of VIDEOPREFER. [†] denotes the realistic videos from existing benchmarks.

| Source | Proportion | Video Length | Type | Resolution |
|---|---|---|---|---|
| LaVie [42] | 16.4% | 2.0s | Diffusion | 512×512 |
| ModelScope [22] | 21.0% | 2.0s | Diffusion | 256×256 |
| VideoCrafter2 [6] | 17.2% | 1.6s | Diffusion | 320×320 |
| Open-Sora [57] | 2.4% | - | DiT | 512×512 |
| Pika[2] | 7.2% | 3.0s | DiT | 1088×640 |
| Text2Video-Zero [16] | 15.3% | 2.0s | Diffusion | 512×512 |
| ZeroScope | 13.8% | - | Diffusion | 1024×576 |
| Gen-2 | 0.7% | - | - | 1792×1024 |
| ActivityNet [5][†] | 1.7% | - | Human-captured | - |
| MSR-VTT [49][†] | 0.03% | - | Human-captured | - |

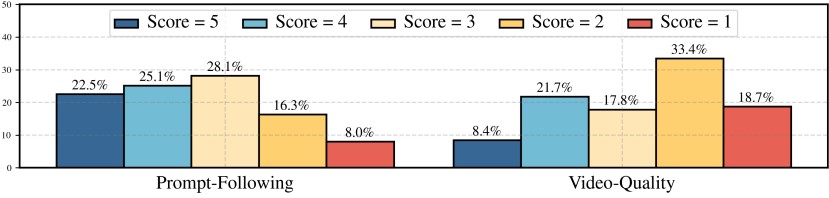

Figure 9: Score distribution across two annotation aspects in VIDEOPREFER.

**Prompt: A man working in his room, typing a portable computer, and a cat watching him**

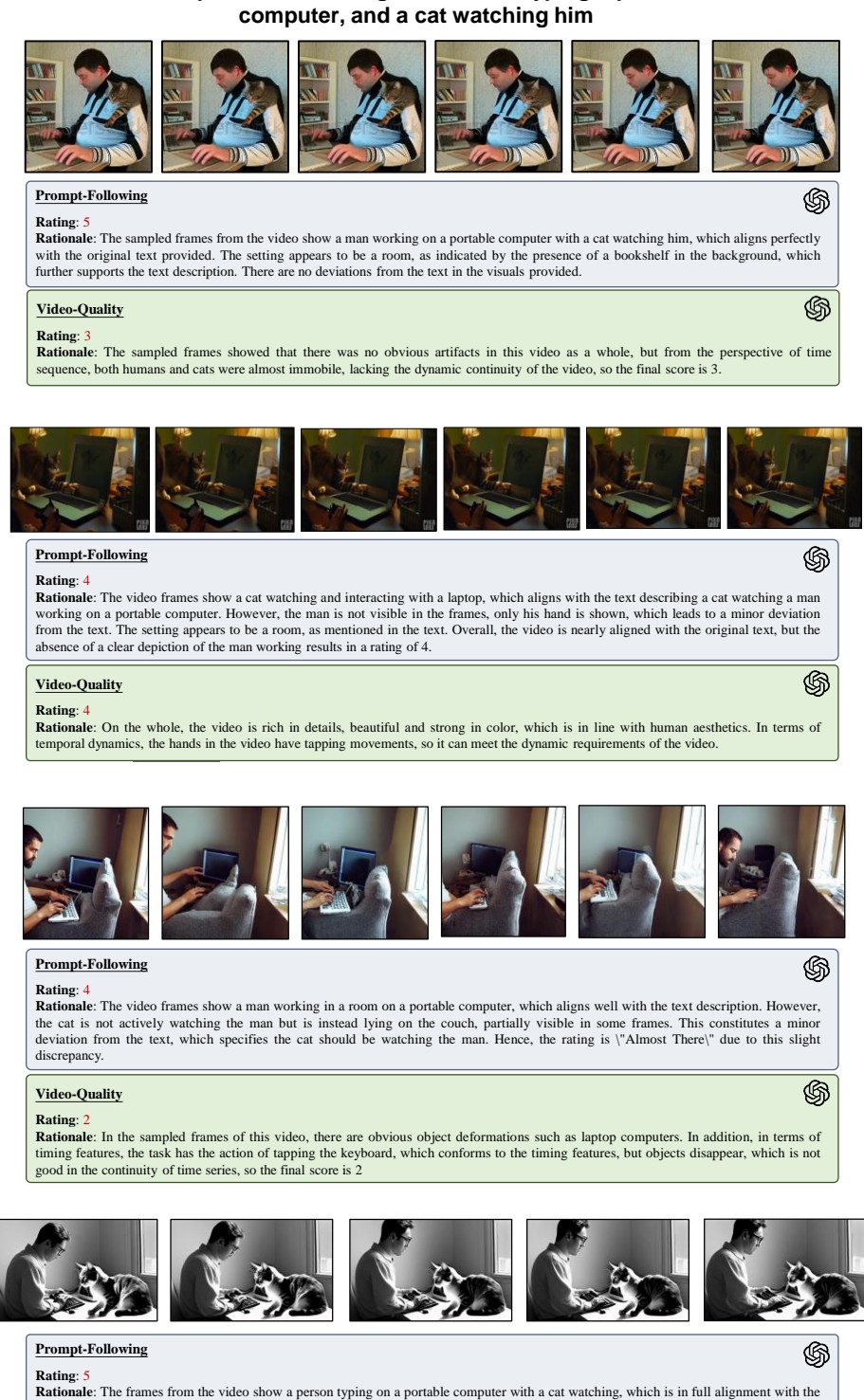

Figure 10: Visualization of example data item in VIDEOPREFER. Here we show one data item which contains a prompt and four corresponding generated videos. For each video, there are two annotations from different annotation aspects (Prompt-Following and Video-Quality) are provided by GPT-4 V

# C Visualization of Selection Results

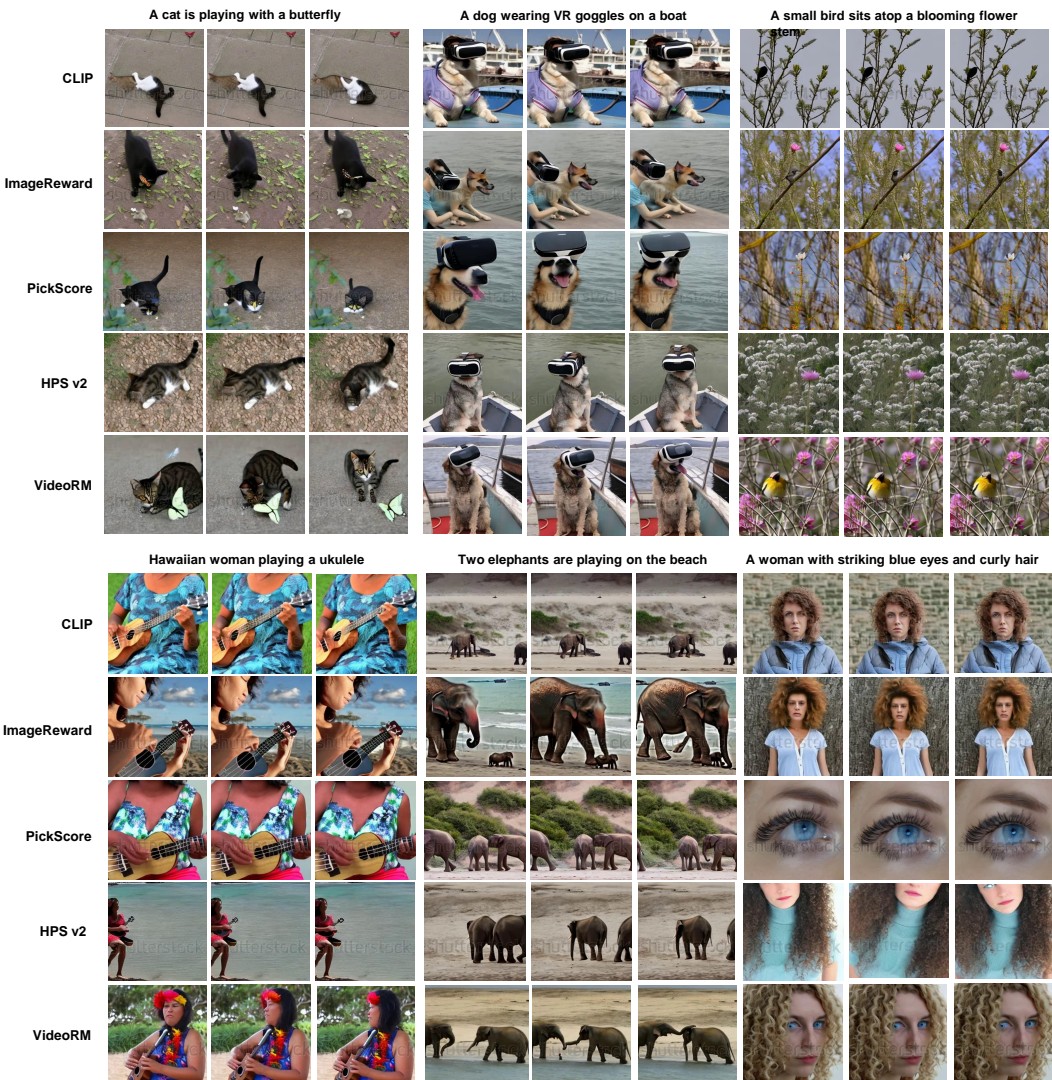

Figure 11: Top-1 videos from 32 generated videos select by CLIP, ImageReward, PickScore, HPS v2 and VIDEORM. **VIDEORM is capable of selecting higher-quality generated videos, e.g., those that better match the prompt descriptions and exhibit more dynamic content.**

# D  Visualization Results for Fine-tuning Text-to-Video Models

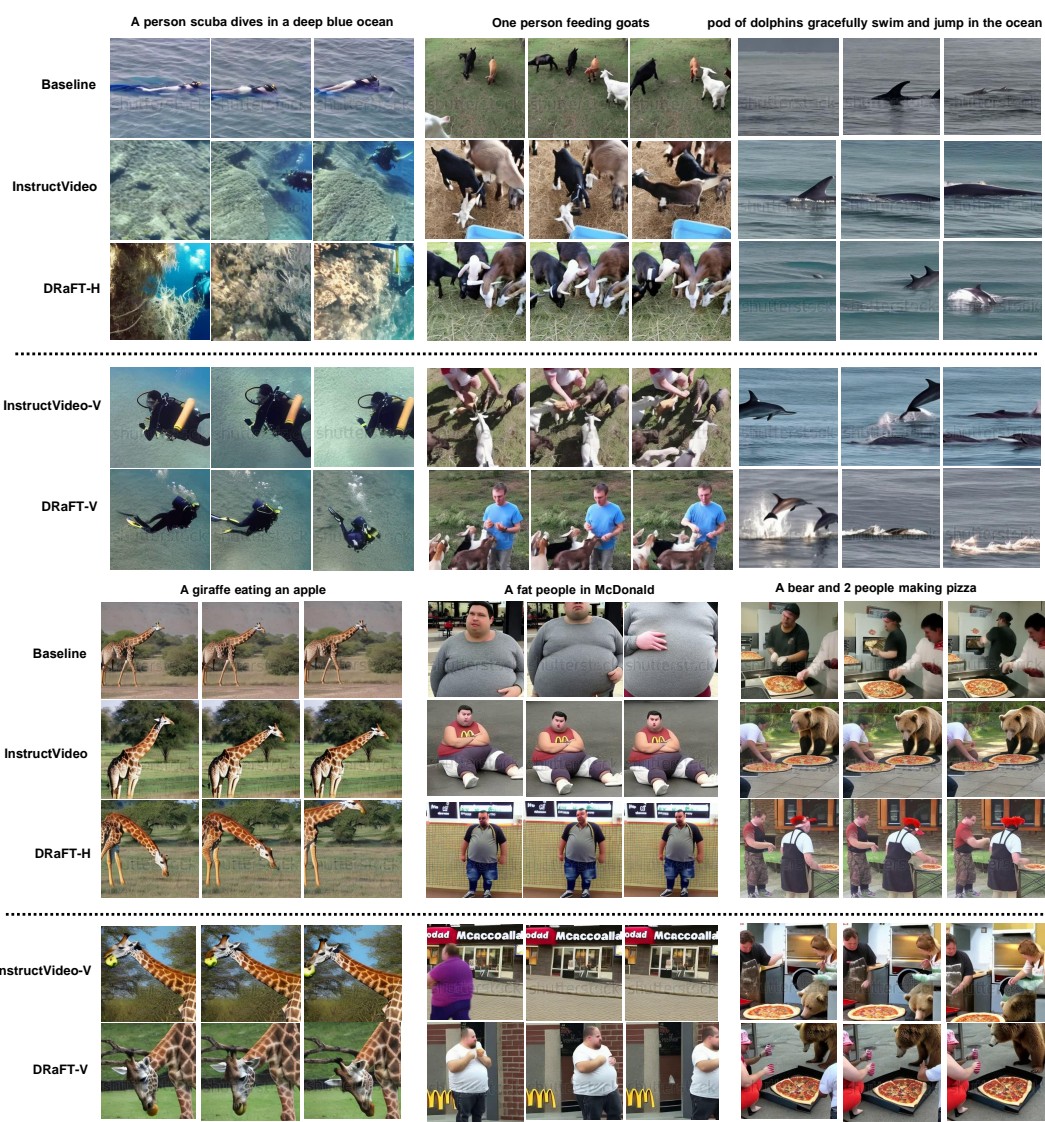

Figure 12: Visualization Results for different fine-tuning methods. We find that compared to fine-tuning with an image domain reward model, fine-tuning with VIDEORM significantly enhances the performance of text-to-video models (DRaFT-V and InstructVideo-V).

# E   Instruction Template

> **Preference Instruction for Prompt-Following**
>
> **Prompt-Following:**
> You are an AI assistant programmed to assess videos with impartial and balanced standards. A video has been created based on a piece of text. Your task is to evaluate how well The content of the video aligns with the original text provided as ("Input") The video evaluation is based on sampled frames shown in sequence.
> **Scoring**: Rating outputs 1 to 5:
>
> 1. **Irrelevant**: No alignment.
>
> 2. **Partial Focus**: Addresses one aspect poorly.
>
> 3. **Partial Compliance**:
>    - (1) Meets goal or restrictions, neglecting other.
>    - (2) Acknowledges both but slight deviations.
>
> 4. **Almost There**: Near alignment, minor deviations.
>
> 5. **Comprehensive Compliance**: Fully aligns, meets all requirements.
>
> Please present your assessment as follows:
> **Output**
> Rating: [Provide the rating]
> Rationale: [Explain the reason for your rating in concise sentences]
>
> ---
>
> Now, review the following video and its corresponding text.
> **Input**:
> #### Text: [INSERT PROMPT HERE]
> #### Frames sampled from video: [INSERT THE FRAMES OF VIDEO HERE]

**Preference Instruction for Video-Quality:**

**Video-Quality:**
You are an AI assistant trained to impartially assess temporal consistency quality, dynamic quality and aesthetic quality in videos. A video has been created based on a piece of text. Your task is to analyze the quality of this video based on the following guidelines and provide a comprehensive evaluation.

**Scoring**: Rating outputs 1 to 5:

1. **Bad**: blurry, underexposed with significant noise, indiscernible subjects, exhibits significant inconsistencies and noticeable discrepancies in appearance of subjects.

2. **Poor**: Noticeable blur, poor lighting, washed-out colors, and awkward composition with cut-off subjects, suffers from noticeable issues in maintaining uniformity of subjects and backgrounds.

3. **Fair**: In focus with adequate lighting, dull colors, decent composition but lacks creativity. Subjects and backgrounds maintain a reasonable degree of uniformity throughout most of the video, with only minor discrepancies.

4. **Good**: Sharp, good exposure, vibrant colors, thoughtful composition with a clear focal point. Good video dynamics and temporal consistency.

5. **Excellent**: Exceptional clarity, perfect exposure, rich colors, masterful composition with emotional impact. perfect temporal consistency and excellent dynamics.

Please present your assessment as follows:
**Output**
Rating: [Provide the rating]
Rationale: [Explain the reason for your rating in concise sentences]
──────────────────────────────────

Now, review the following video and its corresponding text.
**Input**:
#### Text: [INSERT PROMPT HERE]
#### Frames sampled from video: [INSERT THE FRAMES OF VIDEO HERE]

