# OpenReview forum: "Boosting Text-to-Video Generative Model with MLLMs Feedback"
_NeurIPS.cc/2024/Conference — NeurIPS 2024 poster_

### Official Review · Reviewer_cM4S · 2024-07-05

**Soundness:** 3
**Presentation:** 3
**Contribution:** 3
**Rating:** 8
**Confidence:** 4

**Summary:**

Recent text-to-video models like Sora show potential but suffer from poor video quality and misalignment with text prompts due to variable dataset quality. This study addresses these issues by leveraging Reinforcement Learning from Human Feedback (RLHF) to align outputs with human preferences. Due to the high costs of manual annotation, Multimodal Large Language Models (MLLMs) were used for video preference annotations, demonstrating strong concordance with human judgments. This led to the creation of VideoPrefer, containing 135,000 annotations, and the development of VideoRM, a reward model for video preferences. Experiments confirm the effectiveness of both VideoPrefer and VideoRM.

**Strengths:**

Strength:
1. The research focuses on utilizing multimodal large models for dataset annotation, which is a highly promising direction with significant potential for real-world applications. This innovative approach leverages the strengths of multimodal data to enhance the accuracy and efficiency of dataset labeling, offering substantial advancements in the field.


2. The paper introduces a novel reward model in the field of video preference, which effectively evaluates video quality. This approach presents a significant advancement, as it offers a robust method for assessing video content, potentially leading to improved recommendations and enhanced user experience.

3. The quantitative analysis effectively demonstrates the text-to-video selection capability of the proposed VIDEORM model, highlighting its strong semantic alignment abilities. This thorough analysis underscores the model's proficiency in aligning textual inputs with relevant video content, showcasing its potential impact in the field.

**Weaknesses:**

Weakness:

1. Although existing research has demonstrated that GPT-4 can be used for data annotation, it is essential to perform a sample check of the annotated data to assess its quality. Relying solely on the literature to support the use of GPT-4 for annotation without conducting thorough spot checks and corrections undermines the reliability of the annotated content. Ensuring the accuracy and quality of annotations through systematic verification is crucial for maintaining the integrity of the dataset.

2. Furthermore, the evaluation of generated videos should consider perspectives from multiple roles. Incorporating role-playing prompts to capture diverse viewpoints would more accurately reflect the varied opinions and perspectives that different individuals may have regarding the same video. Relying on a single prompt template for evaluation is limiting and does not adequately represent the range of possible reactions and insights.

3. From the content of Algorithm 1's pseudocode, it appears that there is a lack of task-specific algorithmic innovation in the application of reinforcement learning for fine-tuning. The approach seems to follow standard practices without introducing novel techniques tailored to the specific challenges of the task at hand.

4. While the proposed VIDEORM model demonstrates strong performance across various metrics, the paper lacks interpretability experiments. Including such experiments would provide deeper insights into the model's decision-making process and enhance the overall understanding of its effectiveness.

**Questions:**

Weakness:

1. Although existing research has demonstrated that GPT-4 can be used for data annotation, it is essential to perform a sample check of the annotated data to assess its quality. Relying solely on the literature to support the use of GPT-4 for annotation without conducting thorough spot checks and corrections undermines the reliability of the annotated content. Ensuring the accuracy and quality of annotations through systematic verification is crucial for maintaining the integrity of the dataset.

2. Furthermore, the evaluation of generated videos should consider perspectives from multiple roles. Incorporating role-playing prompts to capture diverse viewpoints would more accurately reflect the varied opinions and perspectives that different individuals may have regarding the same video. Relying on a single prompt template for evaluation is limiting and does not adequately represent the range of possible reactions and insights.

3. From the content of Algorithm 1's pseudocode, it appears that there is a lack of task-specific algorithmic innovation in the application of reinforcement learning for fine-tuning. The approach seems to follow standard practices without introducing novel techniques tailored to the specific challenges of the task at hand.

4. Although the proposed VIDEORM model demonstrates strong performance across various metrics, the paper lacks interpretability experiments. Including such experiments would provide deeper insights into the model's decision-making process and enhance the overall understanding of its effectiveness.

**Limitations:**

The preference annotation in this study relies entirely on GPT-4, which is not advisable. While manual annotation is costly, the authors could employ individuals from diverse fields, genders, and age groups to annotate preferences according to a standardized procedure from their unique perspectives. These annotated samples could then be used as prompts for GPT-4 to perform further annotations, potentially improving the quality. Additionally, designing a richer set of prompt templates to capture preferences from multiple perspectives would enhance the dataset's inclusivity and robustness. The current approach lacks the diversity and comprehensive representation necessary for a truly inclusive dataset.

---

> ### Author Rebuttal · Authors · 2024-08-07
>
> Thank you for your detailed feedback. We address your feedback point by point below.
>
> ---
>
> >**Q1**: Although existing research has demonstrated that GPT-4 can be used for data annotation, it is essential to perform a sample check of the annotated data to assess its quality.
>
> **A1**: Thank you for your insightful suggestions. To further validate the reliability of VideoPrefer, we randomly selected 2,000 samples from VideoPrefer and invited six human experts to conduct human preference scoring on two dimensions: prompt-following and video quality. We analyzed the correlation between the human experts' scores and GPT-4V's scores, which are presented in Table A. We found that GPT-4V exhibited excellent alignment with human judgment in both aspects (close to or exceeding 70%). A 70% agreement rate among qualified human preference annotators is a widely recognized benchmark across multiple fields, including preference annotations in NLP[1] and text-to-image[2], further demonstrating the reliability of VideoPrefer.
>
> **Table A.** Correlations between GPT-4 V and human preference judgment across two aspects.
>
> | Prompt-Following | Video-Quality |
> | ---------------- | ------------- |
> | 69.65%           | 73.97%        |
>
> **Reference**
>
> [1] Cui, Ganqu, et al. "Ultrafeedback: Boosting language models with high-quality feedback." ICLR 2024.
>
> [2] Xu, Jiazheng, et al. "Imagereward: Learning and evaluating human preferences for text-to-image generation." NeurIPS 2024.
>
> ---
>
> >**Q2**: Furthermore, the evaluation of generated videos should consider perspectives from multiple roles. Incorporating role-playing prompts to capture diverse viewpoints would more accurately reflect the varied opinions and perspectives that different individuals may have regarding the same video. Relying on a single prompt template for evaluation is limiting and does not adequately represent the range of possible reactions and insights.
>
> **A2**: All the authors agree that your suggestion is very meaningful. We will incorporate your insights into our future research to achieve a more comprehensive evaluation of generated videos. Thank you for your valuable feedback.
>
> ---
>
> >**Q3**: From the content of Algorithm 1's pseudocode, it appears that there is a lack of task-specific algorithmic innovation in the application of reinforcement learning for fine-tuning. The approach seems to follow standard practices without introducing novel techniques tailored to the specific challenges of the task at hand.
>
> **A3**: The primary innovation of this paper lies in demonstrating that MLLMs can provide effective human preference annotation information for text-to-video generation. Based on this, we propose the most comprehensive video preference dataset, VideoPrefer, and the best-performing reward model, VideoRM. Algorithm 1 introduces a framework for fine-tuning text-to-video models with VideoRM, which is **not our main innovation** and indeed lacks task-specific algorithmic innovation. However, **we included it to offer a feasible approach for applying VideoRM to the alignment of text-to-video models**. This framework is **simple** and **effective**, allowing VideoRM to be used for aligning text-to-video models while avoiding the complex application strategies and high computational costs required in previous work[1] for applying image reward models to text-to-video model alignment.
>
> In future research, we will explore and design more novel framework algorithms to achieve better fine-tuning for the alignment of text-to-video models.
>
> **Reference**
>
> [1] Yuan, Hangjie, et al. "InstructVideo: instructing video diffusion models with human feedback." CVPR 2024.
>
> ---
>
> >**Q4**: While the proposed VIDEORM model demonstrates strong performance across various metrics, the paper lacks interpretability experiments. Including such experiments would provide deeper insights into the model's decision-making process and enhance the overall understanding of its effectiveness.
>
> **A4**: Thank you for your interesting suggestion! We will conduct interpretability analyses and experiments on VideoRM in future research to enhance the study's comprehensiveness and completeness, and use these insights to design improved reward models for the text-to-video domain.

---

> > ### Comment · Reviewer_cM4S · 2024-08-09
> >
> > Thank you for your response and detailed explanations. I have updated my rating to 8.

---

### Official Review · Reviewer_qjMG · 2024-07-06

**Soundness:** 3
**Presentation:** 3
**Contribution:** 3
**Rating:** 5
**Confidence:** 4

**Summary:**

Recent advancements in text-to-video generative models, such as Sora, have shown impressive capabilities, generating significant interest for their potential applications. However, these models often rely on extensive datasets of variable quality, resulting in generated videos that may lack aesthetic appeal and fail to accurately reflect the input text prompts. To address this, leveraging Reinforcement Learning from Human Feedback (RLHF) aims to align model outputs with human preferences, but the high costs of manual annotation have limited comprehensive preference datasets. This study investigates the efficacy of Multimodal Large Language Models (MLLMs) for generating annotations, finding a high degree of concordance with human judgments. Building on this, the study uses MLLMs to perform fine-grained video preference annotations, creating VIDEOPREFER, a dataset with 135,000 annotations. Utilizing this dataset, the authors introduce VIDEORM, the first general-purpose reward model for video preference in the text-to-video domain. Comprehensive experiments validate the effectiveness of both VIDEOPREFER and VIDEORM, representing a significant advancement in aligning text-to-video models with human preferences.

**Strengths:**

1. **Advancing Text-to-Video Generative Models with Synthetic Feedback**: This work addresses an important direction in the field by leveraging synthetic feedback to enhance text-to-video generative models. The incorporation of Reinforcement Learning from Human Feedback (RLHF) aims to align model outputs more closely with human preferences, which is crucial for improving the quality and relevance of generated videos. This approach helps overcome the limitations of variable-quality datasets and enhances the models' ability to produce aesthetically appealing and contextually accurate videos.

2. **Significant Contribution through Comprehensive Preference Dataset**: The authors have compiled a preference dataset for text-to-video generative models, consisting of 14,000 prompts, 54,000 videos, and 135,000 preference choices. This dataset, named VIDEOPREFER, represents a significant contribution to the field of text-to-video generation.

3. **VIDEORM Reward Model for Fine-Tuning**: Building on the preference dataset, the authors have trained a reward model named VIDEORM. This model is tailored specifically for video preference in the text-to-video domain and can significantly aid subsequent text-to-video generative models in fine-tuning with high-quality data.

**Weaknesses:**

1. **Representativeness of Generated Video Quality**: Over 98% of the videos in VIDEOPREFER are generated using open-source models (as indicated in Table 5). This high percentage raises a crucial question about the extent to which the quality rankings of these synthetic videos can represent the quality rankings of real videos. It is important to clarify that the concern is not whether GPT-4 V aligns with human judgment, but rather how well the preference dataset and the resulting reward model genuinely capture and represent real-world video quality.

2. **Quality Demonstration through Video Demos for VIDEOPREFER**: The quality of the VIDEOPREFER dataset cannot be fully assessed through the images shown in Figure 10 alone. To better judge the dataset's quality, it is recommended that the authors provide 5-10 videos from the dataset. These demos would offer a clearer and more comprehensive understanding of how well the dataset captures human video preferences and the overall quality of the included videos.

3. **Assessment of VIDEORM through Comparison Demos**: Figure 11 showcases images selected by VIDEORM, but these images alone are insufficient to evaluate the actual quality of the selected videos. To properly assess the effectiveness of the VIDEORM reward model, the authors are recommended to provide 3 comparison sets of the selected videos by different reward models. These demos would allow for a more accurate judgment of the reward model's value and its ability to select high-quality videos that align with human preferences.

4. **Effectiveness of Fine-Tuning with VIDEORM Demonstrated through Demos**: The images in Figure 12 illustrating the results of the VIDEORM fine-tuned model do not adequately convey the actual video quality. To better demonstrate the effectiveness of fine-tuning with the proposed reward model, the authors are recommended to present 3 sets of comparison videos. These demos would offer a concrete demonstration of how fine-tuning with VIDEORM improves video generation quality, thus providing a more tangible proof of the model's effectiveness.

By addressing these points and providing the suggested video demos, the study could offer a more comprehensive and transparent evaluation of the proposed methods and datasets, significantly enhancing the understanding and validation of the results presented.

**Questions:**

Please refer to the weaknesses.

**Limitations:**

This paper discusses the limitations of choosing hyperparameters, which significantly influence the annotation accuracy of GPT-4 V. In addition, it would be better to also discuss the limitations of VIDEOPREFER and VIDEORM.

---

> ### Author Rebuttal · Authors · 2024-08-07
>
> Thank you for your detailed feedback. We address your feedback point by point below.
>
> ---
> >**Q1**: **Representativeness of Generated Video Quality**: Over 98% of the videos in VIDEOPREFER are generated using open-source models (as indicated in Table 5). This high percentage raises a crucial question about the extent to which the quality rankings of these synthetic videos can represent the quality rankings of real videos. It is important to clarify that the concern is not whether GPT-4 V aligns with human judgment, but rather how well the preference dataset and the resulting reward model genuinely capture and represent real-world video quality.
>
> **A1**: Your suggestion is very interesting. In fact, the primary aim of our text is to demonstrate that MLLMs can provide reliable preference annotations for the text-to-video field and to present VideoPrefer, the most comprehensive video preference dataset, as well as the most effective reward model, VideoRM. Therefore, the focus of our research is on improving the quality of synthetic videos (generated by text-to-video models) and their alignment with human preferences. We introduced real videos into VideoPrefer to enhance the dataset's diversity and generalizability.
>
> Moreover, we conducted the following analysis experiment: We randomly selected 500 samples containing real videos from VideoPrefer and conducted a statistical analysis of GPT-4V's preference scores in terms of prompt-following and video quality. We found that the likelihood of real videos receiving high scores was 84.91% and 78.23%, respectively (compared to a random chance of 25%). This indicates that GPT-4V tends to assign higher preference scores to real videos. Additionally, we had six human experts score these samples for preference. We observed that the likelihood of real videos receiving high scores was 82.33% and 88.29%, respectively (compared to a random chance of 25%). This alignment with GPT-4V as the scorer demonstrates that VideoPrefer and VideoRM can effectively capture the high quality of real videos.
>
> ---
>
> >**Q2**: Quality Demonstration through Video Demos for VIDEOPREFER & Assessment of VIDEORM through Comparison Demos & Effectiveness of Fine-Tuning with VIDEORM Demonstrated through Demos.
>
> **A2**: All the authors fully agree with your suggestion to provide more demonstrations comparing VideoPrefer, videos selected by different reward models, and videos generated through fine-tuning with VideoRM. In the latest version of the paper, we will enhance these demos to provide a more robust and effective evaluation and presentation of the quality of our research.
>
> ---
>
> > **Q3**: This paper discusses the limitations of choosing hyperparameters, which significantly influence the annotation accuracy of GPT-4 V. In addition, it would be better to also discuss the limitations of VIDEOPREFER and VIDEORM.
>
> **A3**: Thank you for your suggestions. We will include content about the limitations of VideoPrefer and VideoRM in the limitations section. A concise summary is as follows:
>
> Although VideoPrefer is currently the largest preference dataset in the text-to-video domain, we believe it can be further scaled to achieve better alignment results. In the future, we plan to scale it further. Additionally, we will explore designing more optimal reward model architectures to support more effective video feature modeling, more robust preference prediction capabilities, and more efficient video processing.

---

> > ### Comment · Reviewer_qjMG · 2024-08-12
> >
> > Thank you for the detailed response. While I appreciate the effort to address my concerns, several issues remain that warrant further improvement.
> >
> > Firstly, regarding the representativeness of generated video quality, the analysis provided on the preference scores for real videos is informative, but it does not fully address the core concern. I am not questioning whether GPT-4 V aligns with human judgment, but rather whether the trained VideoRM has the capability to do so. The high percentage of synthetic videos in the dataset may still limit the generalizability of the reward model to practical applications. Scaling up the dataset to include more real video data might make the resulting reward model more robust in guiding the model to generate more realistic videos.
> >
> > Secondly, although the decision to enhance more video demonstrations in a future version of the paper does not help with the current review, I still hope the authors pursue this to further improve the quality of the paper.
> >
> > Lastly, regarding the discussion on hyperparameters, are there any strategies to mitigate their impact? Additionally, the summarized discussion of limitations in the response is appreciated, but it feels somewhat superficial. Could you provide more concrete examples or a deeper exploration of the potential limitations?
> >
> > In conclusion, while the rebuttal addresses some concerns, there remain areas where the paper could be improved. I encourage the authors to consider these points carefully in the revision process.

---

> > > ### Author Response · Authors · 2024-08-13
> > > **Response to Reviewer qjMG**
> > >
> > > Thank you for providing us with detailed feedback. We have greatly benefited from your insights. Below are our responses:
> > >
> > > >**Q1**: I am not questioning whether GPT-4 V aligns with human judgment, but rather whether the trained VideoRM has the capability to do so. The high percentage of synthetic videos in the dataset may still limit the generalizability of the reward model to practical applications. Scaling up the dataset to include more real video data might make the resulting reward model more robust in guiding the model to generate more realistic videos.
> > >
> > > **A1**: All authors agree with your suggestions. We plan to release the second version of VideoPrefer in the future. In this upcoming version, one of our primary improvements will be to increase the proportion of real videos in the dataset, thereby making the reward model more robust. We outline the specific steps as follows:
> > >
> > > **Step 1**: Use well-established event detection models (e.g., [mmaction2](https://github.com/open-mmlab/mmaction2) to segment existing large-scale real video datasets (such as [Kinetics-700](https://paperswithcode.com/dataset/kinetics-700)) into multiple smaller video clips.
> > >
> > > **Step 2**: Generate captions for the obtained real video clips using a mature and reliable large model, e.g.,  GPT-4V.
> > >
> > > **Step 3**: Input the generated captions into randomly selected video generation models to produce corresponding synthetic videos.
> > >
> > > **Step 4**: Combine the captions, real video clips, and the generated synthetic videos to create new VideoPrefer data examples.
> > >
> > > By implementing these steps, we will be able to increase the proportion of real videos in the dataset.
> > >
> > > ---
> > >
> > > >**Q2**: Secondly, although the decision to enhance more video demonstrations in a future version of the paper does not help with the current review, I still hope the authors pursue this to further improve the quality of the paper.
> > >
> > > **A2**: Thank you for the reminder. We will certainly include additional video demonstrations across various aspects (such as the assessment of VideoRM or the results of fine-tuning) in the latest version of the paper, as per your request, to further enhance the quality of the paper and improve the visualization of the dataset and experimental results.
> > >
> > > ---
> > >
> > > >**Q3**: regarding the discussion on hyperparameters, are there any strategies to mitigate their impact?
> > >
> > > **A3**: One approach we have considered is to mitigate the impact of hyperparameter selection by conducting multiple annotations for each sample using GPT-4V. Specifically, we would annotate each sample N times, with the hyperparameters randomly sampled for each annotation. The final preference score for the sample would be the average of these N scores. We believe this method could help reduce the influence of hyperparameter choices to some extent. Of course, this is a very interesting research problem in itself, and we will continue exploring possible solutions to further improve the accuracy of the annotation.
> > >
> > > ---
> > > >**Q4**:  Additionally, the summarized discussion of limitations in the response is appreciated, but it feels somewhat superficial. Could you provide more concrete examples or a deeper exploration of the potential limitations?
> > >
> > > **A4**: Firstly, inspired by your feedback, we recognize a potential limitation of our dataset: the relatively low proportion of real videos, which might affect the robustness of the trained VideoRM (a specific solution for this can be found in **A1**). Another possible limitation we have identified is the lack of a comprehensive bias assessment for the VideoPrefer dataset in our current research. For instance, it remains unclear whether GPT-4V exhibits a noticeable preference for certain types of videos. Although we have analyzed the preference between real and synthetic videos within VideoPrefer and found that GPT-4V shows a strong preference for real videos (93% vs. 70%), further detailed analysis is needed. This includes assessing whether there is a preference for videos generated by specific models or for videos of certain styles. We plan to conduct more in-depth analyses in future research to address these potential limitations.
> > >
> > > ---
> > >
> > > **If you have any further questions or concerns, please feel free to contact us at any time. We are always available and look forward to further discussions with you. :)**
> > >
> > > Best regards,
> > >
> > > All Authors

---

> > > > ### Comment · Reviewer_qjMG · 2024-08-14
> > > >
> > > > Thank you for the further discussion and the detailed responses to the queries raised. At this point, I have no additional questions. I am confident that by incorporating these new discussions and additional experimental results, the quality of the final version will be significantly enhanced. I look forward to seeing the effects in the final version. Best of luck!

---

### Official Review · Reviewer_snrb · 2024-07-09

**Soundness:** 3
**Presentation:** 2
**Contribution:** 3
**Rating:** 6
**Confidence:** 4

**Summary:**

The paper presents a dataset and a learned reward model to better align the generated outputs of text-to-video models with human preferences. Specifically, the authors make use of GPT-4V to provide preference scores for a large dataset of videos by learning from human feedback on a smaller set of videos. The authors then develop their proposed architecture for the reward model on top of this dataset to predict preference scores given prompts and videos. They fine-tune text-to-video models with their proposed reward model and show the improvements in alignment with human feedback through quantitative evaluations, ablation experiments, qualitative results, and user studies.

**Strengths:**

1. The proposed idea of leveraging an LLM such as GPT-4V to create a large-scale synthetic dataset for the text-to-video paradigm is useful. It makes good use of existing technology to explore previously intractable problems.

2. The reward model design is technically sound and rigorous.

3. The experiments presented in the paper are extensive, covering multiple LLMs for dataset creation, multiple feature representations for text-to-video models, and multiple datasets for showing model performances.

**Weaknesses:**

1. Some key aspects of the dataset collection process are unclear.

    1.1. While GPT-4V has the best agreement with human feedback, it is still not very high (around 70%, according to the authors). Did the authors explore any prompt-tuning or other approaches to improve the agreement? Did the authors observe any potential patterns in the disagreement? Perhaps certain categories of videos have more agreement between GPT-4V and human feedback than other categories?

    1.2. When collecting data from video caption datasets, do the authors process the captions in some way to make them structurally and semantically similar to video generation prompts?

    1.3. Did the authors run any data filtering for discriminatory, harmful, or abusive content in the collected/generated prompts and videos?

    1.4. Did the authors consider various levels of complexity in the prompts, such as foreground and background details, number of objects, actions, interactions, etc.?

    1.5. How did the authors quality-check the videos generated for the proposed dataset? It is not fully clear how the proposed approach avoids a cyclic process, where improving video generation depends on a better reward model, which, in turn, requires generated videos of a certain quality.

    1.6. Can the authors please clarify the specific numbers regarding the dataset size? How did they go from 14K data items to 135K preference choices?

2. Some key aspects of the user study are unclear, making it hard to fully follow the results.

    2.1. Is five participants a sufficiently high number from which to draw conclusions? How many videos did each participant respond to? Was there a mix of non-experts and subject matter experts such as content creators or video editors? If so, were their responses segregated in any way?

    2.2. Did the authors check for participants' engagement, e.g., whether participants responded too quickly or too slowly, and whether any such responses needed to be discounted? Did the authors check for any recency bias or response drifts, where the participants may have responded differently to similar video content over time?

**Questions:**

1. In their ablation study, why do the authors consider up to 12 frames? Is this limit due to computational constraints?

**Limitations:**

Yes, but needs more details on how the authors filter for discriminatory, harmful, or abusive content in their collected dataset.

---

> ### Author Rebuttal · Authors · 2024-08-07
>
> Thank you for your feedback. Due to space constraints, we address your questions in the **Rebuttal** below as well as at the top of this page in the **Author Rebuttal**:
>
> ---
> >**Q1**: While GPT-4V has the best agreement with human feedback, it is still not very high. Did the authors explore any prompt-tuning or other approaches to improve the agreement? Did the authors observe any potential patterns in the disagreement?
>
> **A1**: **A 70% agreement rate among qualified human preference annotators is a widely recognized benchmark across multiple fields, including preference annotations in NLP[1] and text-to-image[2].** This means that when different human experts provide preference annotations for the same sample, their agreement rate is typically around 70%. With GPT-4V achieving an agreement rate of 69.65%, we consider it a reliable preference annotator in the text-to-video domain.
>
> We explored the impact of different input templates and hyperparameters (as discussed in Section 5) on the accuracy of GPT-4V annotations and identified the most suitable input template and hyperparameters. We believe that exploring prompt-tuning or other approaches to improve agreement is a very interesting research topic, and we plan to explore this further in future research.
>
> Additionally, our analysis shows that in VideoPrefer samples containing real videos, the agreement between GPT-4V and human feedback is higher, reaching approximately 93% (compared to about 70% in samples without real videos). Both GPT-4V and human feedback tend to assign higher preference scores to real videos in these samples, indicating that the quality of videos generated by existing models still needs improvement compared to real videos.
>
> **Reference**
>
> [1] Cui, Ganqu, et al. "Ultrafeedback: Boosting language models with high-quality feedback." ICLR 2024.
>
> [2] Xu, Jiazheng, et al. "Imagereward: Learning and evaluating human preferences for text-to-image generation." NeurIPS 2024.
>
> ---
> >**Q2**: Do the authors process the captions to make them similar to video generation prompts?
>
> **A2**: We did not apply any special processing to the captions; instead, we used them directly as input prompts for the video generation model to generate videos. We believe this approach ensures that samples in VideoPrefer, composed of real and generated videos, fairly correspond to the same prompt. If we were to preprocess the prompts, we are concerned that differences between prompts corresponding to generated and real videos might introduce potential bias into the final preference results.
>
> ---
> >**Q3**: Did the authors run any data filtering for prompts and videos? Needs more details on how the authors filter for discriminatory, harmful, or abusive content in their collected dataset.
>
> **A3**: The prompts in VideoPrefer are primarily sourced from the VidProM dataset, as well as the MSR-VTT and ActivityNet video caption datasets. The VidProM dataset has already implemented harmful content filtering to screen prompts (consisting of toxicity, obscenity, identity attack, etc.). Since MSR-VTT and ActivityNet are widely used video caption datasets released some time ago, we believe they contain minimal harmful information. Therefore, we did not perform additional harmful content filtering on the prompts. Similarly, the videos in VideoPrefer are mainly sourced from some open-source video generation models and video caption datasets, and we have not applied data filtering to them. Your suggestion is valuable, and we will incorporate harmful content filtering for both prompts and videos in the v2 version of VideoPrefer to ensure the dataset is more legally compliant and appropriate.
>
> ---
> >**Q4**: Did the authors consider various levels of complexity in the prompts?
>
> **A4**: In this study, VideoPrefer directly uses the existing prompts from the VidProM dataset, as well as the MSR-VTT and ActivityNet video caption datasets, without considering complexity levels. Your suggestion is valuable :), and we plan to distinguish and categorize the complexity levels of prompts in future research. We will make the necessary adjustments to enhance the effectiveness of VideoPrefer.
>
> ---
> >**Q5**: How did the authors quality-check the videos generated for the proposed dataset? It is not fully clear how the proposed approach avoids a cyclic process, where improving video generation depends on a better reward model, which, in turn, requires generated videos of a certain quality.
>
> **A5**: **The quality check for the generated videos primarily relies on GPT-4V, i.e., MLLMs**. VideoPrefer is essentially a preference dataset in the text-to-video domain, where each sample includes a prompt and four video candidates corresponding to that prompt. GPT-4V scores these four videos on two dimensions: prompt-following and video quality (see Section 2.2). The comparison of preference scores between high-scoring and low-scoring videos helps us train an effective reward model (see Section 3.2) to assess the degree of human preference for videos.
>
> >**Q6**: Can the authors clarify the the dataset size? How did they go from 14K data items to 135K preference choices?
>
> **A6**: Our VideoPrefer dataset contains 14k prompts, each with four video candidates. These four video candidates are scored by GPT-4V on two dimensions: prompt-following and video quality. This allows us to perform pairwise comparisons across these two dimensions, allowing for 12 pairwise preference comparisons per prompt. Therefore we obtain 14k × 12 = 168k preference choices. After filtering out comparisons where the preference scores were equal (as these cannot be used to optimize the reward model), we finalized 135k preference choices.
>
> ---
> >**Q7**: Are five participants sufficient? How many videos did each participant respond to? ...?
>
> >**Q8**: Did the authors check for participants' engagement? ...?
>
> >**Q9**: In their ablation study, why do the authors consider up to 12 frames?
>
> **A**: Pleasee see in the **Author Rebuttal**.

---

> > ### Comment · Reviewer_snrb · 2024-08-13
> > **Response to Authors**
> >
> > I thank the authors for their detailed rebuttal, which addresses all my concerns. I maintain my original recommendation for acceptance and have slightly raised my score.

---

### Official Review · Reviewer_nZ7n · 2024-07-12

**Soundness:** 3
**Presentation:** 2
**Contribution:** 2
**Rating:** 5
**Confidence:** 4

**Summary:**

This paper introduces a new dataset called VideoPrefer, a collection of (simulated) human preferences on videos conditioned on certain language prompts. VideoPrefer utilizes GPT-4v as an automatic reward assessor, which contains videos from both machine-generated and real-world curated videos. The work then utilizes the collected preference as a reward to learn a reward model named VideoRM, constructed on top of a prior model HPS v2, extending its capabilities to temporal dimension and thus can present reward modeling in videos.

**Strengths:**

- This work collects a large scale of preference data, potentially useful for evaluating text-to-video generative models.
- It is demonstrated that the designed reward model VideoRM is showing the most aligned preferences over a few baseline methods.
- Finetuned text-to-video models utilizing the presented VideoRM seems effective under the designed experimental settings.

**Weaknesses:**

- While I appreciate the Table 2’s analysis on TVGE dataset among the tested video assessor candidates, solely performing statistical analysis on the VideoPrefer dataset is still required. People would be relying on the VideoRM which is trained on the VideoPrefer data and it remains questionable how faithful the dataset is.
- Lack of some analysis between real-world and generated videos. Under the same prompt, are real-world videos always or more likely to be better? What heuristics can people use to expand beyond the collected video sets?
- Is it unclear how text prompts are aligned with real-world videos in Section 2.2.
- Lack of rigorous statistical analysis on the curated video prompts. How diverse are they? What is the type-token ratio? What are the top frequent predicates and affected entities? What are the genres? It is hard to understand how challenging and/or how underlying bias would have affected the video generation process from this curated resource.
- Why only consider extension of HPS v2 as the base reward model? There are plenty of good video-language models that can be potentially useful (such as [1]) to be adapted as a reward model. If the work can show more unbiased learning from the dataset solely with video-based preference models, it would strengthen the work more.

[1] Sung, Yi-Lin, Jaemin Cho, and Mohit Bansal. "Vl-adapter: Parameter-efficient transfer learning for vision-and-language tasks." CVPR 2022.

**Questions:**

- There are some minor typos, for example, L84 the “K” is already a thousand there. Please be more mindful when writing.
- What or who is deciding the “win” in Figure 2? The GPT-4v assessor?

**Limitations:**

- The limitations of this work do not seem to be explicitly addressed.

---

> ### Author Rebuttal · Authors · 2024-08-06
>
> Thank you for your feedback. Due to space constraints, we address your questions in the **Rebuttal** below as well as at the top of this page in the **Author Rebuttal**:
>
> ---
> > **Q1**: Solely performing statistical analysis on the VideoPrefer dataset is still required. It remains questionable how faithful the dataset is.
>
> **A1**: Thank you for your insightful suggestions. To further validate the reliability of VideoPrefer, we randomly selected 2,000 samples from VideoPrefer and invited six human experts to conduct human preference scoring on two dimensions: prompt-following and video quality. We analyzed the correlation between the human experts' scores and GPT-4V's scores, which are presented in Table A. We found that GPT-4V exhibited excellent alignment with human judgment in both aspects (close to or exceeding 70%), further demonstrating the reliability of VideoPrefer.
>
> Additionally, we provide a comprehensive statistical analysis and presentation of VideoPrefer in Section B of the appendix, including the distribution of preference scores, the distribution of video sources, and more. In the latest version of the paper, we will also include more detailed statistical analyses of VideoPrefer, such as the score distribution between real and generated videos, statistical analysis of the prompts collection, and bias analysis. These will serve as references for those using VideoPrefer.
>
> **Table A.** Correlations between GPT-4 V and human preference judgment.
>
> | Prompt-Following | Video-Quality |
> |-|-|
> | 69.65%|73.97%|
>
> ---
> >**Q2**: Under the same prompt, are real-world videos always or more likely to be better? What heuristics can people use to expand beyond the collected video sets?
>
> **A2**: We conducted the following analysis: We randomly selected 500 samples containing real videos from VideoPrefer and performed a statistical analysis of GPT-4V's preference scores on prompt-following and video quality. We found that real videos had a high likelihood of receiving top scores, at 84.91% and 78.23%, respectively (compared to 25% in a random scenario). This indicates that GPT-4V tends to assign higher preference scores to real videos. To explore whether this phenomenon is reasonable, we asked six human experts to also score these samples. We found that the likelihood of real videos receiving high scores was 82.33% and 88.29% in prompt-following and video quality, respectively (25% in a random scenario). This indicates that real videos generally receive higher preference scores, also highlights that the current quality of video generation models still needs improvement.
>
> Additionally, this inspired us to consider generating more comparison videos using video generation models for each sample containing real videos and using real videos as cases that typically receive higher human preference scores to rapidly expand the preference dataset. We will further explore the effectiveness and potential limitations of this method in future research.
>
> ---
> >**Q3**: Is it unclear how text prompts are aligned with real-world videos in Section 2.2.
>
> **A3**: In fact, we use the current video caption dataset directly as the source for obtaining text prompts and corresponding real video pairs. Specifically, for each prompt in the video caption dataset, we generate three additional video candidates using a randomly sampled generative model. These generated video candidates, along with the original prompt and the corresponding real video clip, together constitute a single sample of VideoPrefer.
>
> ---
> > **Q4**: Why only consider extension of HPS v2 as the base reward model? There are plenty of good video-language models that can be potentially useful to be adapted as a reward model.
>
> **A4**: The reasons for selecting the extension of HPS v2 as the base reward model are as follows:
>
> 1. **High Performance and Low Bias**: HPS v2 is the best-performing reward model trained on the largest debiased preference dataset in the text-to-image domain, with minimal bias. Since video preference scores are highly correlated with the preference scores of each frame in the video, HPS v2 naturally serves as an effective reward model for the video generation domain[1]. By using HPS v2 as the initial model, we provide our reward model with a strong initial point and foundational knowledge, thus enhancing the effectiveness of the reward model.
> 2. **Efficient Deployment**: During the alignment process of video generation models, the deployment cost of the reward model (in terms of parameters, computational load, and inference time) can significantly impact the alignment performance under resource constraints. HPS v2 has a much lower deployment cost compared to some video-language models, making it easier to deploy during the model alignment process.
>
> We also finetuned the video-language models you recommended[2] (denoted as VL-Model) on both open-source human-crafted preference datasets and VideoPrefer, using the same training steps & data as VideoRM. The preference prediction accuracy is shown in Table B. We found that its performance was not as good as VideoRM, which further demonstrates the effectiveness of using HPS v2 from the text-to-image domain as the foundation for our reward model.
>
> **Table B.**
>
> |Model|TVGE|VBench|T2VQA-DB|
> |-|-|-|-|
> |HPSv2|69.5|55.7|52.8|
> |VideoRM|**73.7**|**63.5**|**65.4**|
> |VL-Model|64.44|53.38|53.09|
>
> **Reference**
>
> [1] Yuan, Hangjie, et al. "InstructVideo: instructing video diffusion models with human feedback." CVPR 2024.
>
> [2] Sung, Yi-Lin, et al. "Vl-adapter: Parameter-efficient transfer learning for vision-and-language tasks." CVPR 2022.
>
> > **Q5**: There are some minor typos.
>
> **A5**: We apologize for the typos and will correct them in the latest version of the paper.
>
> ---
> > **Q6**: What or who is deciding the “win” in Figure 2?
>
> **A6**: Pleasee see in the **Author Rebuttal**.
>
> ---
> >**Q7**: The limitations do not seem to be explicitly addressed.
>
> **A7**: Pleasee see in the **Author Rebuttal**.

---

> > ### Comment · Reviewer_nZ7n · 2024-08-12
> >
> > Thanks for the responses.
> > Some of the points addressed my doubts, however, the original manuscript was a bit far from publication ready.
> > I retain my score.

---

> > > ### Author Response · Authors · 2024-08-13
> > > **Thanks**
> > >
> > > Dear Reviewer nZ7n
> > >
> > > We would like to sincerely thank you for your feedback and suggestions :). We will carefully revise our paper, addressing all issues (e.g., typos and adding statistical analysis on the curated video prompts) to make the paper clearer and more complete.
> > >
> > > Best regards,
> > >
> > > All Authors

---

### Author Rebuttal · Authors · 2024-08-06

## Supplementary rebuttal for Reviewer nZ7n

---

> **Q6**: What or who is deciding the “win” in Figure 2? The GPT-4v assessor?

**A6**: The "win" in Figure 2 is determined by scores from six human experts. The sample with the highest average expert score is considered the ground truth. We apologize for omitting this explanation in the paper, and we will include it in the revised version.

>**Q7**: The limitations do not seem to be explicitly addressed.

**A7**: In Section 5 and Figure 8 of our paper, we have conducted extensive comparative ablation experiments and analyses for choosing hyperparameters. We hope this can serve as a reference for future similar works when selecting hyperparameters. A relatively efficient solution, in our opinion, is to first use different hyperparameters to generate small-scale MLLM annotations and assess their quality. Once the best hyperparameter combination is identified, large-scale annotation generation can proceed (this is the method we used to construct VideoPrefer).

We will include this limitation as a separate section in the updated version of our paper. We also believe that finding efficient and effective methods for choosing hyperparameters in MLLM annotation generation is a very interesting research direction, and we plan to conduct further studies in this area.


##
## Supplementary rebuttal for Reviewer snrb

---
>**Q7**: Is five participants a sufficiently high number from which to draw conclusions? How many videos did each participant respond to? Was there a mix of non-experts and subject matter experts such as content creators or video editors? If so, were their responses segregated in any way?

**A7**: **Based on previous related work, where three participants were used for human studies in the text-to-image domain[1] and the NLP domain[2], we believe that having five participants rate the videos is sufficient to support our conclusions.** Each participant rated all the results, and every generated video used for evaluation was scored by all participants. The final score for each video was determined by the average score from all participants.

Among the five participants, there were three experts (in image aesthetics, video aesthetics, and image quality) and two non-experts to ensure a more comprehensive and accurate evaluation. During the evaluation process, the five participants did not engage in any form of discussion or communication. Thank you for your question, and we will include these details in the updated version of the paper.

**Reference**

[1] Xu, Jiazheng, et al. "Imagereward: Learning and evaluating human preferences for text-to-image generation." NeurIPS 2024.

[2] Cui, Ganqu, et al. "Ultrafeedback: Boosting language models with high-quality feedback." ICLR 2024.

---

>**Q8**: Did the authors check for participants' engagement, e.g., whether participants responded too quickly or too slowly, and whether any such responses needed to be discounted? Did the authors check for any recency bias or response drifts, where the participants may have responded differently to similar video content over time?

**A8**: We did not check the time each participant took to respond to ensure they didn't respond too quickly or too slowly. However, we implemented a **Post-Labeling Check** mechanism for each participant to ensure the reliability of the final results. Participants were required to review 20% of randomly selected samples for quality checking and rescore them. If the scores differed by more than 25%, that participant's score for the video would not be considered.

---

>**Q9**: In their ablation study, why do the authors consider up to 12 frames? Is this limit due to computational constraints?

**A9**: We considered using up to 12 frames based on the conventional settings from previous works that employed models with clip-like structures for video-related tasks[1,2]. Subsequently, we increased the number of frames to 24 and 32 for further implementation. The experimental results are presented in Table A, where we observed that the performance of VideoRM improved with an increased number of frames. However, considering the deployment cost and speed of aligning video generation models, as well as the performance of VideoRM, we determined that using 8 frames is the most suitable choice.

**Table A. Pair-wise preference prediction at TVGE dataset for VideoRM when using different input frames.**

| frames                  | 4    | 8    | 12   | 24   | 32   |
| ----------------------- | ---- | ---- | ---- | ---- | ---- |
| **prediction accuracy** | 72.8 | 73.7 | 73.5 | 74.0 | 74.4 |

**Reference**

[1] Luo, Huaishao, et al. "Clip4clip: An empirical study of clip for end to end video clip retrieval and captioning." Neurocomputing 2022.

[2] Wang, Mengmeng, et al. "Actionclip: Adapting language-image pretrained models for video action recognition." TNNLS 2023.

---

### Decision · Program_Chairs · 2024-09-25

**Decision:**

Accept (poster)

**Comment:**

The paper presents a dataset and a learned reward model for video preference to help align text2video generation models. Using GPT-4V, authors automatically label a large dataset of videos to learn this preference model, and fine-tune text-to-video models with this proposed preference model showing improvements in alignment with human feedback through quantitative/human evaluations and ablation experiments.

The paper was reviewed by four expert reviewers, all of who leaned positively towards accepting the paper. ACs concur with the reviewer consensus, and believe this is an important field of study for further improvements in text2video models. Authors must incorporate additional discussions and experimental results shared/promised during the rebuttal phase into their final paper.